# Visualization of the Dynamic Brain Activation Pattern during a Decision-Making Task

**DOI:** 10.3390/brainsci12111468

**Published:** 2022-10-29

**Authors:** Harshit Parmar, Eric Walden

**Affiliations:** Texas Tech Neuroimaging Institute, Texas Tech University, 2500 Broadway, Lubbock, TX 79409, USA

**Keywords:** decision making, dynamic activation, fMRI, visualization, deconvolution

## Abstract

Decision making is a complex process involving various parts of the brain which are active during different times. It is challenging to measure externally the exact instant when any given region becomes active during the decision-making process. Here, we propose the development and validation of an algorithm to extract and visualize the dynamic functional brain activation information from the observed fMRI data. We propose the use of a regularized deconvolution model to simultaneously map various activation regions within the brain and track how different activation regions changes with time, thus providing both spatial and temporal brain activation information. The proposed technique was validated using simulated data and then applied to a simple decision-making task for identification of various brain regions involved in different stages of decision making. Using the results of the dynamic activation for the decision-making task, we were able to identify key brain regions involved in some of the phases of decision making. The visualization aspect of the algorithm allows us to actually see the flow of activation (and deactivation) in the form of a motion picture. The dynamic estimate may aid in understanding the causality of activation between various brain regions in a better way in future fMRI brain studies.

## 1. Introduction

The use of functional magnetic resonance imaging (fMRI) to study the neuroscientific underpinnings of decision making has been quite popular and successful. Studies have revealed various factors that affect decision making such as reward [1,2], risk [3,4], uncertainty [5,6], and morality [7,8]. These studies have relied on examining static differences in a contrast model, thereby averaging decision making across time. However, decision making is a dynamic process [9]. Unfortunately, it is challenging to design an analysis pipeline to handle continuous processes within the GLM (generalized linear model) framework. 

There are numerous issues that need to be addressed before any inference can be made from the raw fMRI time series. First, the useful blood oxygenation level-dependent (BOLD) signal is only 2–5% of the absolute intensity [10,11]. Second, because the fMRI measures the indirect effect of neuronal activity, the neuronal signal is convolved with the hemodynamic response and is shifted in time by 4–6 s [12,13]. Third, the fMRI time series is very noisy. A typical fMRI time series is confounded with various noise sources including physiological artifacts such as cardiac and respiratory noise, motion artifact, system induced noise such as gaussian and thermal noise, signal drift due to scanner instabilities and background noise [14,15]. 

In traditional task fMRI data analysis, the predicted BOLD response is obtained by convolving the experiment design with the hemodynamic response function [16]. In decision-making studies, it is difficult to determine the exact onset time of individual steps involved in the decision-making process, thus it is challenging to achieve an accurate experiment design that can be used to obtain a reliable predicted response. Moreover, the entire decision making is a continuous process and different parts of the brain are involved at different time instances [9,17]. Here, we propose the use of a deconvolution-based technique for estimation and visualization of continuous dynamic brain activity during a simple binary decision-making task. Instead of estimating the predicted response, the observed BOLD time series is used to obtain an estimate of the experiment paradigm for each voxel, henceforth referred as the ‘pseudo-stimulus’. It is assumed that the pseudo-stimulus causes the observed BOLD signal changes, and thus can be an accurate representation of ongoing dynamic brain activity. The use of a hemodynamic response model corrects for hemodynamic delays observed in the BOLD signal and the inference from the pseudo-stimulus can be time locked to the actual experiment design. Regularization is also used to reduce the effects of noise. It is shown in the Appendix B section that the shape of the extracted pseudo-stimulus remains close to the simulated ground truth even for low-SNR observed signals.

Deconvolution models have been used for analysis of fMRI data to extract the hemodynamic response function [18,19,20,21] and its variability [22]. Moreover, deconvolution models have also been used to identify underlying neural events from the observed BOLD time series [23,24,25]. Commonly, hemodynamic deconvolution models use lasso regularization (L1 regularization), which results in temporally sparse neuronal activity [26,27,28]. Among the main goals was to obtain a continuous visualization of the decision-making process while the output of L1 regularization was sparse. The decision-making task used in this study is also a block design and not event related to making a continuous interpretation more useful. A variant of L1 regularization, L1 regularization of the first difference of the weight vector, was also tested (described in Appendix B section) and the results of that were similar to L2 regularization. Thus, L2 regularization [29] was used instead, which provides a smoother (temporally) more continuous output with a shorter computation time. The output of L1 regularization is also more susceptible to noise as compared to L2 regularization. Moreover, the solution to the L2 regularized cost function can be obtained using a closed-form equation while the solution to L1 regularized cost function is obtained via an iterative process making L1 regularization a time- and resource-intensive process [30]. Because of the requirements for the application and faster computation times, L2 regularization was preferred over the standard L1 regularization.

The main contribution of this article is the development and validation of a model-free approach that can estimate and visualize the voxel-wise dynamic brain activation (and deactivation) without any knowledge about the original experiment design model. The proposed technique is tested on simulated dataset with varying parameters and then applied to a real decision-making task. The brain activation during the entire decision-making task can be visualized using a motion picture and various regions involved in the decision-making process are highlighted. Another unique aspect about this study is the use of a multi-band pulse sequence which allows for faster temporal sampling. The fast sampling also allows for the acquisition of more information and better estimation of the underlying causal neuronal process.

Section 2 below describes the proposed algorithm, the experimental setup, and the entire processing pipeline. Section 3 shows the results first on simulated data and then on real fMRI data for a binary decision-making task followed by inference and discussion of the results. Additional information about algorithm validation, hyperparameter tuning and comparison with other similar technique is shown in the Appendix B sections. 

## 2. Materials and Methods

### 2.1. Algorithm

A typical task fMRI analysis uses the GLM framework. Among the important steps in the GLM framework is the creation of a design matrix. A design matrix is generated using the experiment timing information. For decision-making tasks, the timing for external experiment parameters can be measured but it is practically not possible to measure the exact instance of decision making while the participant is still inside the scanner. Instead of specifying known event onset and offset times, the current approach instead creates a hypothetical HRF for each time point and then tests to see which HRFs fit the data that are observed. At each time point, a question is asked—would an HRF that started here give rise to data such as the data being observed? We can then find times at which an HRF in a particular voxel would have generated data such as the actual data. Thus, we do not need the onset and offset times of a known stimuli. Instead, we can induce the onset and offset times of a stimuli that would give rise to the data from the observed data itself. We call this a pseudo-stimulus. We assume this pseudo-stimulus to be an input as perceived by different parts of the brain. The proposed algorithm tries to estimate the pseudo-stimulus for each voxel of the brain based on the observed BOLD fMRI data. 

We first explain the algorithm using a single time series and then later extended to time series of all the voxels within the brain. The entire algorithm is shown in Figure 1. The most common step in a task fMRI analysis is the generation of an estimated response time series. The estimated response is obtained by convolving the experiment design with the HRF. In Figure 1, the experiment design is shown in part (C), typical hemodynamic response is shown in part (D) and the estimated response time series is shown in black in part (E) of Figure 1. In this study, we are interested in obtaining the experiment design from the actual voxel time series. The most straight forward way of doing that is to use a deconvolution model with the observed time series and the HRF model. However, the deconvolution models are very sensitive to noise. Given the fact that the fMRI time series are very noisy, outputs of the deconvolution models are not reliable. Thus, a different approach must be used to estimate the experiment design from the observed BOLD signal.

The observed signal is assumed to be a weighted sum of individual hemodynamic responses at different time instances. The colored plots in part (E) of Figure 1 show how the predicted response can be obtained as weighted sums of hemodynamic responses at different time instances. Each colored plot shows the hemodynamic response at different time instances. The first step in the approach is to form the design matrix. The design matrix consists of 3 main parts. First, an impulse function is generated for each TR (repetition time) (part A). This is convolved with an HRF model to form a series of hemodynamic responses that would have been generated by a pseudo-stimulus at every TR in this study (part B). 

Next, a set of cosine basis function which corresponds to the low-frequency signal drift is added to the matrix. The number of cosine basis sets can vary depending upon the overall experiment paradigm and the total duration of fMRI acquisition. The low-frequency fluctuations are generally of the order of one cycle ever 100–150 s. The smallest frequency in the cosine basis set corresponds to a half cycle in entire scan duration and the remaining frequencies are integer multiples of it. The sampling frequency of the cosine basis should be set to be 1/TR Hz. Finally, the last part consists of a constant term which accounts for offset and absolute baseline intensity of the voxel. The fMRI data are preprocessed and corrected for motion and temporal signal drift. The cosine basis set and the constant terms are only to account for any residual signal bias or drift (if any) in the preprocessed data.

The linear model used for estimation of the pseudo-stimulus is specified in Equation (1), where Y corresponds to the observed signal, DM corresponds to the design matrix, W corresponds to the weights for the regressors (pseudo-stimulus) and ε corresponds to unmodelled noise. In Equation (1) are DM ϵ R^T × N^ (N = T + Nc + 1; T is the number of TR in fMRI data; Nc is the number of cosine basis sets), Y ϵ R^T × Nv^ (Nv is the number of voxels inside the brain), and W ϵ R^N × Nv^. There are more features than data points, so L2 regularization, also known as ridge regression [29], is used to estimate a set of weights, where the observed activation levels are the dependent variable. The fMRI time series is noisy and, with the help of L2 regularization the pseudo-stimulus, can be estimated with decent accuracy even at low SNR (discussed in Appendix B section). Equation (2) shows the optimization equation used for approximation of W, where ‘λ’ is the regularization constant and ‘I’ is an identity matrix of size ‘N’. Typically, for fMRI studies, a different λ value is used for each voxel but here a fixed value of λ is used. The selection of λ value and why a constant value is used for λ are discussed in the Appendix B section. The closed-form solution for Equation (2), which is used for software implementation, is shown in Equation (3).
(1)Y~ DM×W+ε
(2)$$L\left(w\right)=\frac{1}{2N}\;\overset{N}{\underset{n=1}{\sum }}{\left({\hat{Y}}_{n}-Y_{n}\right)}^{2}\;+\frac{\lambda }{2}\;\overset{N}{\underset{n=1}{\sum }}\|w_{n}{}^{2}\|$$
(3)$$\underset{W}{\mathrm{argmin}}\;L\left(w\right)\sim \;{\left(\frac{DM^{T}\cdot DM}{N}+\lambda I\right)}^{-1}\left(\frac{DM^{T}}{N}\cdot Y\right)$$

The size of the estimated pseudo-stimulus matrix W will be N × Nv, where each column corresponds to the estimated weights for a single voxel. The pseudo-stimulus for each voxel can be obtained from a subset of W including first T rows and all Nv columns. Each row in the subset of W can be converted to a 3D volume, corresponding to the size of the brain, and volumes from consecutive rows corresponds to consecutive time points (TRs). All T consecutive weight volumes can also be compiled into a single motion picture to visualize the dynamic process going on inside the brain.

### 2.2. Experiment Protocol

The experiment was a simple binary decision-making task, where the participants had to decide whether they would download an app or not. A total of 50 different mobile applications were selected form the Google Play android app store. The apps were relatively uncommon, and a post-scan survey was conducted to know how many of the apps were previously used by the participants. On average participants had used only 3 out of 50 apps, with 8 being the maximum.

When inside the scanner, anatomical images were obtained first followed by the functional scans. The task was to look at the name, logo, screenshot and a short description of the apps and decide whether to download the app or not. The responses were indicated by a button press with the right index finger for YES and right middle finger for NO. As soon as the participant presses the button, there is a blank screen with a small crosshair in the middle for 5 s followed by the details of the next app. The blanking period is to allow for the blood flow to return to baseline before the beginning of the next stimulus. The app order was randomized for each participant. The description of the apps was character matched so that descriptions for all the apps were between 270 and 280 characters. Additionally, all the descriptions were fed to a text to speech (TTS) software and the reading time of the TTS software for all the apps was between 17 and 18 s. Considering the 17 s time to read the text, responses were only valid 10 s after the beginning of the stimulus. The participants could respond before 10 s, but it does not move to the next app and those responses (trials) were not considered for analysis. Thus, a button pressed after 10 s initiated a change to the next app. The 10 s window allowed discarding participants who did not perform the trial seriously. There was no upper bound time limit, and the participants can take as long as they want to respond to a particular app. Figure 2 shows the experiment protocol for a single app.

A total of 22 undergraduate students from a large Western university participated in this study. The participants received bonus course credits for their participation. There were 9 males and 12 females (one participant choose not to specify their gender) with an average age of the population being 20.86 ± 1.75 years (min = 19, max = 26). 21 out of 22 participants were right hand dominant. The participants were asked for fill out an informed consent and an MRI safety screening form prior to being in the MRI scanner. Institutional review board (IRB) approval was also obtained for the experiment. The participants were also given instructions about the task and practiced the task on a laptop outside the scanner. The practice task had the exact same user interface, but the apps used in the practice were different from the apps used in actual experiment.

The MR scans were performed on a 3T Siemens Skyra scanner. The anatomical scans were acquired using a MPRAGE pulse sequence with repetition time (TR) of 1900 ms and an echo time (TE) of 2.49 ms. Anatomical scans consist of 192 sagittal images (each 0.9 mm thick) with an in-plane resolution of 1 mm × 1 mm. The functional scan was acquired using a multi-band echoplanar imaging (MB-EPI) pulse sequence [31], with a TR of 545 ms and a TE of 29 ms. Each functional volume was acquired with 48 axial slices (6 slices acquired at once because of multi-band) of thickness 3.3 mm and in-plane resolution of 3.25 mm × 3.25 mm. As the participants responded at their own pace for each stimulus, the total scan time, and thus the total number of functional volumes acquired varied from 1496 (~15 min) to 3129 (~29 min).

### 2.3. fMRI Data Analysis

The preprocessing is performed using SPM 12 toolbox [32] on MATLAB 2020a. The preprocessing includes

For motion correction, rigid body affine transformation is applied to all the functional volumes to align them to the first functional volume. The rigid body transformation accounts for 6 degrees of freedom that includes 3 translation and 3 rotational motions.After motion correction, the functional and anatomical volumes are coregistered. A 3D affine transform with 12 degrees of freedom is used to align the anatomical and mean functional volume.Next, the images are normalized by mapping both anatomical and functional volumes onto the MNI152 brain atlas [33]. The functional volumes were mapped onto the atlas with a spatial resolution of 3 mm × 3 mm × 3 mm while the anatomical volumes were mapped onto the atlas with spatial resolution of 1 mm × 1 mm × 1 mm.Temporal signal drift was reduced and spatial smoothing was applied to all the functional volumes. Signal drift was estimated and reduced using a principal components analysis-based technique [34]. The spatial smoothing was performed using a 3D Gaussian kernel with a full width half maximum (FWHM) of 5 mm.The normalized volumes are then segmented into gray matter, white matter, cerebrospinal fluid (CSF), skull and skin. The segmented gray matter, white matter and CSF volumes are used to obtain a brain mask. Any voxels outside the brain region were discarded from further analysis to reduce the amount of data and the computation time.

Before explaining each component of the estimation, we will lay out the overall process.

Extract the pseudo-stimulus for each voxel.Normalize the response times.Average together all of a single participants response level across all trials.Repeat for only the trials with a yes and only the trials with a no answer.Use clustering to compress the 50,000+ voxel responses into 20 clusters.

The preprocessed data are then used to extract the pseudo-stimulus for each participant. First, the entire 4D fMRI volume is converted to a 2D matrix of size T × Nv. Nv is the total number of voxels that lie within the brain region and T is total number of brain volumes in the data. Each column of this matrix corresponds to that voxel’s time series. This is the matrix ‘Y’ as described in Section 2.1. From the matrix ‘Y’, the weight matrix is estimated, which is of the size N × Nv, where N corresponds to the number of regressors in the design matrix. The first ‘T’ rows of the weight matrix correspond to the pseudo-stimulus estimate for each of the Nv voxels. The voxel-wise pseudo-stimulus is estimated for all the participants.

The response times varied across apps and across participants, thus a one-to-one comparison is not possible. All the stimuli where participants responded before 10 s were discarded. The average number of stimuli that were discarded per participant was 5.3 (min = 0; max = 16; median = 4). Thus, it can be assumed that all the responses were at least 10 s long and with 5 s blanking period, giving a total time of 15 s. To achieve uniformity in analysis, the onset of the stimulus is considered as the reference point and 13 s (25 TRs, including the reference point) before and after the reference point are considered for each app. Thus, for each stimulus, a fixed-length window is obtained, which is centered at the onset of the stimulus. This fixed window allows us to compare the response across different apps and participants. For each participant, the pseudo-stimulus in the fixed window across all the apps is averaged together to obtain a participant-level response to the stimulus.

For each participant, the pseudo-stimulus is used to obtain the participant-level response to the app download decision. The participant-level response matrix is of size 49 x Nv (the reference point being the 25th row). Let us assume that the kth trial began at time instance tk. Corresponding to that, the time window would be [tk − (24) • TR] to [tk + (24) • TR]; for k = {1, 2, 3, …, 50}. For a given participant, all the 49 x Nv matrices (corresponding to each value of k) are averaged together to obtain single participant-level response to the app.

The participant-level response shows how the brain activity of the participant changed when responding to the app download decision. Three separate responses were obtained for each participant, one for all the apps, one for apps for which the participant decided to download (YES apps) and the last one for the apps the participant did not decide to download (NO apps).

Finally, considering the responses for each voxel as features, the voxels are clustered into 20 different groups using the k-means clustering approach. For each cluster, a representative response is obtained by grouping the voxel response for all the voxels belonging to same cluster. At the end of clustering, the 49 × Nv matrix is converted to a 49 × 20 matrix, with each column containing the temporal response to a single cluster.

The clustering and representative cluster response were obtained for all participants. However, the cluster assignment for k-means clustering is randomized and a correlation-based post-processing step was used to match the clusters across all participants. The group-level analysis was performed by combining the spatial clusters and cluster response for all participants. Useful information can be extracted from the spatial clusters and cluster-wise response.

## 3. Results and Discussion

The modified deconvolution algorithm was first tested on a synthetic data with single time series. Figure 3 shows the results for simulated data. Figure 3a shows the actual experiment design used to generate a synthetic time series and the estimated pseudo-stimulus. The actual experiment stimulus was convolved with the hemodynamic response function to generate the synthetic BOLD time series. Figure 3b shows the clean (black dotted) and noisy (grey) synthetic BOLD time series along with the reconstructed time series (blue) as estimated by the proposed algorithm. The noisy time series consists of thermal noise and signal drift components added to the clean signal. The estimated signal drift/baseline is also shown in Figure 3b in red. From Figure 3a, it can be observed that the estimated pseudo-stimulus closely follows the actual stimulus. The Pearson correlation coefficient is computed to quantify the similarity between the actual and the estimated stimulus. The correlation between the actual and the estimated stimulus is 0.9126. Another interesting thing to observe is that the short-duration stimuli are estimated with a relatively lower amplitude and longer duration. A potential explanation for that could be the saturation effect of the sum of individual HRF. For example, if the amplitude of the estimated response saturates after summing over six HRFs, then any stimulus lasting smaller than six TRs will have a relatively smaller amplitude than other stimuli which are longer than six TRs. Thus, one limitation that can be identified here is the low sensitivity of the approach in detecting short-duration stimuli when present along with longer-duration stimuli.

The proposed algorithm was used to estimate the pseudo-stimulus for all the voxels of the brain while performing a simple binary decision-making task. The estimated pseudo-stimulus for all the voxels can be visualized in the form of a motion picture where each frame corresponds to a single time instance. The visualization of the pseudo-stimulus for the brain responding to the app download decision is shown in Appendix A. The animation shows the average response across all participants. The relative time is shown on top. The instance when the screenshot for a new app appears (beginning of new trial) is considered as T = 0. The entire visualization if from 25 TRs (13.5 s) before the reference point to 25 TRs after the reference point. The 25 TRs after the reference point show the initial process of information gathering and early decision making. The 25 TRs before the reference point captures the events happening during the final decision-making process and the blanking period between two stimuli. The button press happens 5 s before the reference point. Thus, anything happening before that can be considered to be a part of the final decision-making task. The brain activity between the reference time point and 5 s before it indicates what happens after the end of the stimulus during rest period. The pseudo-stimulus amplitude is normalized and converted to z-score. The amplitude corresponds to the strength of activation at any given time instance (estimated from the actual BOLD fMRI time series). The activation strength is corrected for the hemodynamic response delay and thus can be time locked with the experiment paradigm. From the animation a flow of brain activity can be observed in some of the regions, especially the visual cortex regions. The activity starts to peak with the onset of the stimulus, it peaks for a while and then it drops below baseline for some time (deactivation) before reaching the baseline again towards the end of resting period. Different brain regions being activated at the same time can be visualized simultaneously.

The temporal response for voxels in different brain regions is shown in Figure 4. At first, the temporal response may seem very similar to the hemodynamic response obtained using the FIR modeling of the fMRI data [35] but there is a fundamental difference here. The hemodynamic response represents the response of the brain to a given stimulus. The hemodynamic response is obtained by averaging sections of the actual BOLD response and can be considered as the output of the brain to a stimulus. The temporal response or the pseudo-stimulus, on the other hand, represents the perceived input to the brain (can be different for different regions). An experiment paradigm is usually considered as an input to the brain but not all brain regions respond to it at the same time. Some brain regions may receive an indirect input signal from other brain regions which directly respond to the external experiment paradigm. The input to such indirect regions would be different from the experiment paradigm. Thus, the input to all brain regions may not be the same and varying input to different parts of the brain is estimated in the form of a pseudo-stimulus or a temporal response. By knowing the estimated input to different brain regions, it can be possible to identify at what time instance a given brain region was triggered. Apart from that, the hemodynamic response is convoluted with the HRF and is delayed by a few seconds w.r.t to the actual stimulus [12], which is not the case with the pseudo-stimulus. The algorithm corrects for HRF delays, and the temporal response can be time locked with the experiment paradigm providing a better temporal understanding of the dynamic processing of the brain.

Coming back to Figure 4, the light-colored lines represent the response for each participant while the solid black line represents the averaged response. The temporal response is for all the apps irrespective of the download decision by the user. The time instance of button-press and the beginning of the stimulus is indicated with vertical dotted lines. The beginning of the stimulus initiates strong activation in the visual cortex region. The visual cortex activation is caused due to the appearance of the screenshot and app description after the blank screen. The algorithm does correct for the hemodynamic response delays, and thus the temporal response can be time locked to the experiment paradigm. Once the screenshot is displayed, the participants are likely to engage in the task of reading the description of the apps. The reading activity causes a gradual and prolonged activation of the Lingual gyrus which is involved in identification and recognition of words [36]. During the same time a prolonged activation is also observed in the Broca and the Wernicke language regions. The activation of this region can be associated with semantic and syntactic interpretation of the app description [37,38,39]. Towards the end of the stimulus, decision-independent activation is observed in the anterior cingulate cortex (ACC). ACC is shown to be involved in decision-making tasks, especially outcome evaluation before an actual decision is made [40,41,42]. Finally, strong activation is observed in the left motor cortex just before the button press. Little to no activation is observed in the right motor cortex during the same time. The left motor cortex activation is presumably caused due to the finger movement with the right hand to indicate the download response.

Finally, during the blanking period between two stimuli, activation is observed in the default mode network. The default mode network has been shown to be a task-negative brain region meaning it gets deactivated during the task and gets activated in the absence of any specific task [43]. An interesting temporal response is observed for the voxels within frontoparietal network. The temporal response shows a transient behavior where the activation peaks near button press and beginning of the new stimulus. The transient behavior of the temporal response may indicate the involvement of the region in task switching. Previous studies have suggested the role of frontoparietal network as a flexibility hub [44] and in task switching [45]. Appendix A and Figure 4 show the application of the proposed approach in extracting the dynamic spatial and temporal response to the app download decision-making task.

The temporal response from some of the brain regions depended on the user response for the apps. To identify the regions that responded differently for download and not download decisions, a paired t-test was performed. The algorithm extracts the pseudo-stimulus for all the apps whose download decision was either ‘yes’ or ‘no’. A paired t-test was performed on temporal response of each voxel and all participants. Only two brain region showed statistically significant difference in temporal response between ‘yes’ and ‘no’ response, the right ventrolateral prefrontal cortex (VLPFC) and the ventromedial prefrontal cortex (VMPFC). The VMPFC region has been shown to be involved in decision making involving reward [2,46,47]. Figure 5 shows the temporal response for the voxel showing the maximum difference in each region. The response is shown for both ‘yes’ and ‘no’ apps and for all participants. It can be observed from the figure that there is more activation for ‘no’ apps as compared to ‘yes’ apps, right before the final decision time.

The k-means clustering results are summarized in Figure 6. The figure shows spatial clusters and the corresponding temporal response for the cluster. The cluster numbers are randomly assigned by the clustering algorithm and does not relate to the anything related to the stimulus. The cluster-level temporal response is obtained from the centroids of the k-means. Temporal response and spatial parcellation for all the 20 clusters are shown in the Appendix B section. The color scheme corresponds to the user response to the apps. The pseudo-stimulus for all the voxels were extracted for three different cases. First, for all the apps, irrespective of their download decision; second, for all the apps where the user decided to download the apps; and third, for all the apps where user decided not to download the app. The spatial cluster color and the time series color for each of the three cases are also shown in the figure. For spatial clusters, if there is an overlap in cluster for more than one cases (which was common), then the cluster color is obtained by mixing the individual colors. The k-means clustering was able to identify different brain regions based on their response to the stimuli. The activation peaks in the temporal response for different clusters corresponds to different stages in the decision-making task. The clustering approach was able to identify different brain regions in an unsupervised fashion. For example, clusters 14 and 16 correspond to the visual cortex, cluster 2 corresponds to the lingual gyrus, while cluster 18 contains the Broca and Wernicke language regions. All the above-mentioned clusters are active after the start of the stimulus during the initial phase of information gathering.

Before the decision is made (indicated by a button press), activity peaks for cluster 5. Cluster 5 consists of spatial regions belonging to the executive control network, ACC, and dorsolateral prefrontal cortex (DLPFC). DLPFC has been shown to be active towards the end of the decision-making process [5,9,17]. Because of the similar temporal response, all the different spatial regions are grouped into a single cluster. Cluster 13 consists of the left motor cortex region while cluster 8 corresponds to the DMN. Motor cortex shows activity at the end of the stimulus while the DMN shows activity during the 5 s blanking period. Finally, cluster 3 overlaps with the frontoparietal network and shows transient activity during the beginning and end of the main stimulus. The role of DMN and frontoparietal network have been discussed earlier. For clustering, it is interesting to observe that even with an unsupervised clustering approach, it is possible to extract spatial regions with meaningful interpretations.

The proposed technique shows promising results in identification and visualization of the dynamic brain behavior. The extracted brain regions and their temporal activation can be used to infer the role of the given region during the decision-making process, which can be justified by previously published literature. Among the main advantages offered by the proposed technique is the detection of the pseudo-stimulus without the use of the experiment design. Due to that, the technique can be easily applied to a complicated task- and event-related experiment design with varying trial durations. Another advantage of the proposed technique over other deconvolution approach is the ability to extract the pseudo-stimulus for entire brain (50,000+ voxels) in a non-iterative single-step approach. Other approaches use an iterative approach, and the computation time can reach up to days for single subject analysis [30]. A simple time requirement calculation is shown in the Appendix B section.

However, the algorithm is still in its early development stage and can be improved further. The results for a binary decision-making task suggest that regularized deconvolution methods can be used to extract and visualize whole brain dynamic activation information from noisy fMRI data. There are some challenges that exist for effective operation of the proposed approach. First, there are a few hyperparameters in the algorithm and the best combination of the hyperparameters can still be explored. The set of hyperparameters used for obtaining the results shown in this paper were obtained from testing on synthetic and simulated data. The hyperparameter that has the maximum effect on the estimation of the pseudo-stimulus is the regularization constant. Setting it too small may result in a noisy estimate while setting it too large may reduce the resolving ability for smaller stimuli. Some effects of the regularization constant for the closed-form solution have been tested and described in the Appendix B section. Some of the earlier studies have shown SNR-based regularized constant value selection techniques, mainly targeted for iterative L1 regularization [28].

As discussed earlier, another limitation is the inability to fully resolve smaller (short time) stimuli due to the saturation effect. Because of its sparse nature, an L1 regularization approach may provide superior results for identification of neuronal activation in an event-related experiment design. One more limitation lies in the selection of the HRF. It has been tested that small variations in the amplitude and time shifting do not have a large-scale effect on the estimation of the pseudo-stimulus. The details of the HRF variability have been discussed in the Appendix B section. However, large variation in the HRF may result in inaccurate results. Another possible future work can be to use and check the effects of region-specific HRF models for estimation of the pseudo-stimulus.

## 4. Conclusions

The main objective of this paper is to present a technique that can be used to estimate and visualize the dynamic nature of whole-brain activity. The proposed technique is based on the linear deconvolution model. The technique was applied to a simple binary decision-making task and various regions involved in the decision-making process were identified. The focus was more on the approach itself rather than the decision-making inferences, and thus the discussion about the neurological interpretation is limited. However, a detailed analysis was conducted on the algorithm itself. From the results, it is clear that the proposed algorithm is capable of extracting the region-level stimuli as seen by the brain. The so-called pseudo-stimuli is also visualized in the form of a motion picture which clearly shows dynamic brain activity across the whole brain. With that, and a little post-processing, meaningful spatial clusters can also be obtained in an unsupervised manner by using the extracted pseudo-stimuli. There are many aspects of the algorithm that can be improved but the identification of whole-brain activity information can be very useful for future research and in better understanding the dynamic process happening inside the brain during various tasks.

## Figures and Tables

**Figure 1 brainsci-12-01468-f001:**
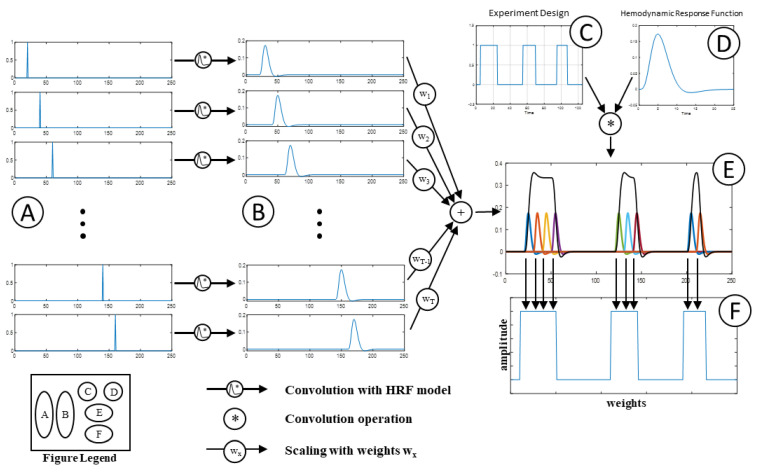
Summary of the algorithm for extraction of the pseudo-stimulus from given time series. (**A**) Train of impulses delayed in time. The time delay between each pair of impulse is equal to 1 TR. (**B**) The impulse train convolved with the HRF to give time-shifted hemodynamic responses. The time-shifted hemodynamic responses are used as regressors in the design matrix. (**C**) Typical fMRI experiment design. (**D**) Standard double-gamma hemodynamic response function (HRF). (**E**) Observed time series shown as the weighted sum of time-shifted hemodynamic responses. (**F**) The extracted pseudo-stimulus, obtained from the weights of the time-shifted hemodynamic responses.

**Figure 2 brainsci-12-01468-f002:**
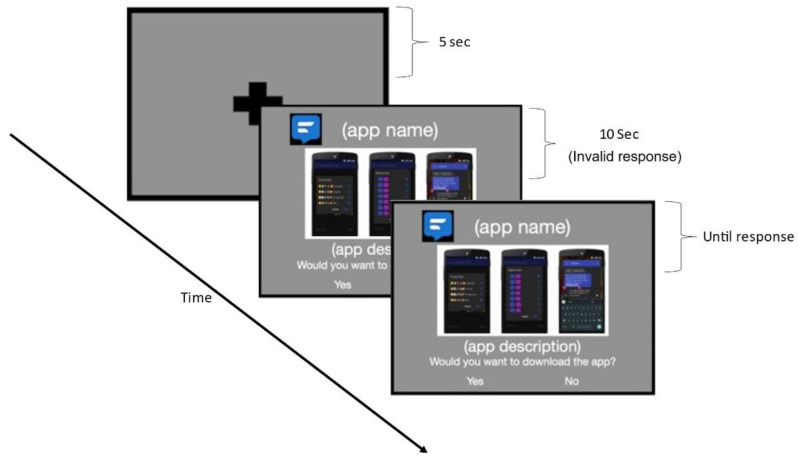
Representation of the experiment protocol. There is a 5 s blank screen before the beginning of the stimulus. After the stimulus starts, the response during the first 10 s is not considered valid. After 10 s, the response by the user is considered valid and it will end the stimulus to bring back the blank screen before beginning the next stimulus.

**Figure 3 brainsci-12-01468-f003:**
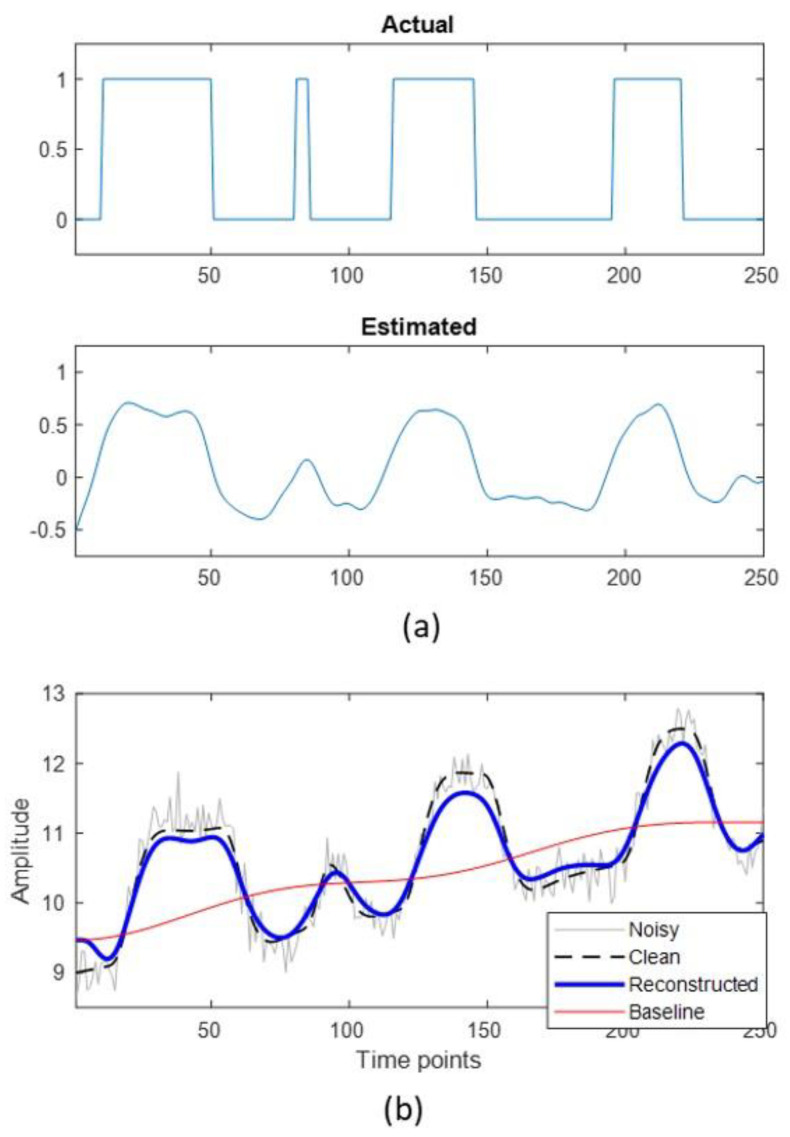
(**a**) The actual (**top**) and the estimated (**bottom**) stimulus for a synthetic time series data. (**b**) synthetic time series. Noisy time series was obtained by adding gaussian noise and signal drift to the clean time series. The noisy time series was used as input for the proposed approach to obtain the estimated time series and estimated signal drift (baseline).

**Figure 4 brainsci-12-01468-f004:**
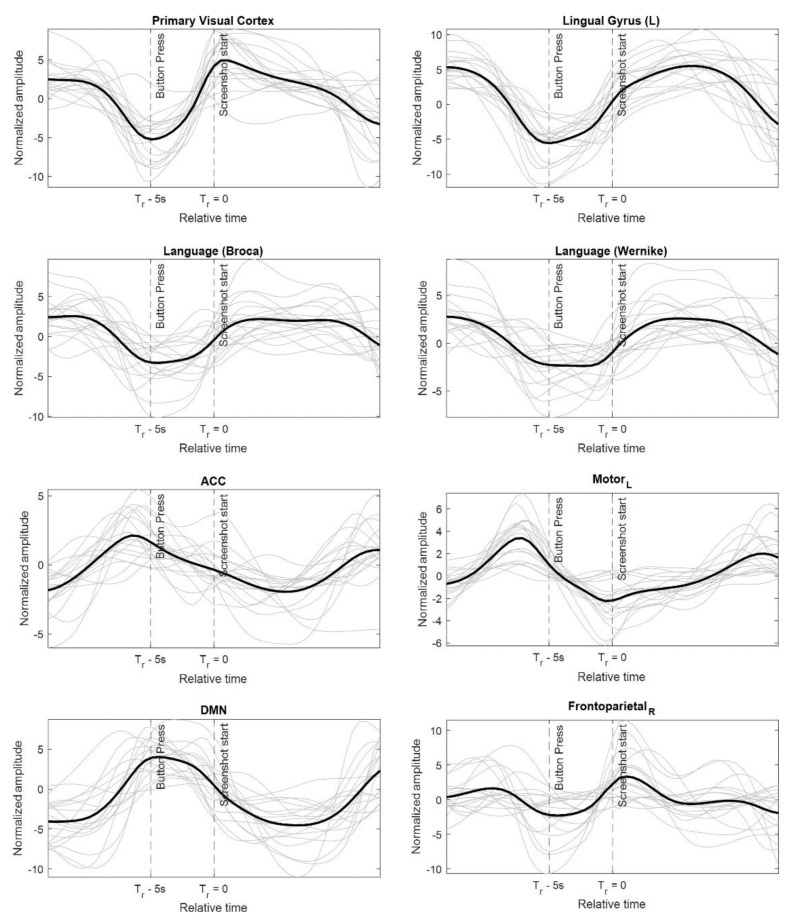
The estimated pseudo-stimulus (temporal response) for different voxels inside the brain. The plots in light grey correspond to the temporal response from single subjects while the dark black plot is the average across all participants. The vertical dotted line indicates the time instance of button press and beginning of the stimulus.

**Figure 5 brainsci-12-01468-f005:**
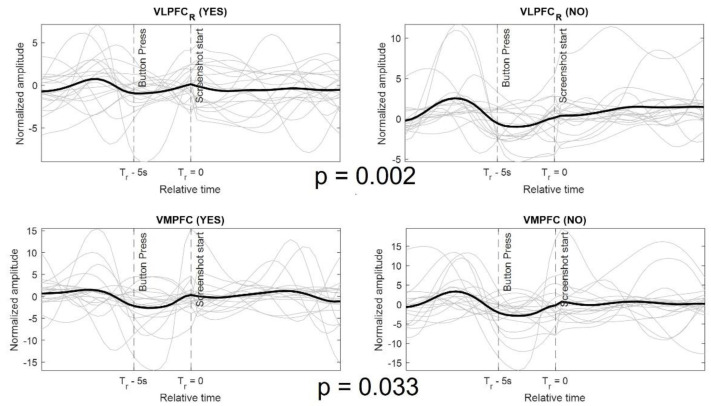
Difference in the pseudo-stimulus for ‘yes’ (**left column**) and ‘no’ (**right column**) download decisions by the participants. The temporal response is shown for voxels in two regions that show a statistically significant difference between the ‘yes’ and ‘no’ apps. The *p*-values for the paired t test are shown between the subplots. The color coding of the plots is same as in Figure 4.

**Figure 6 brainsci-12-01468-f006:**
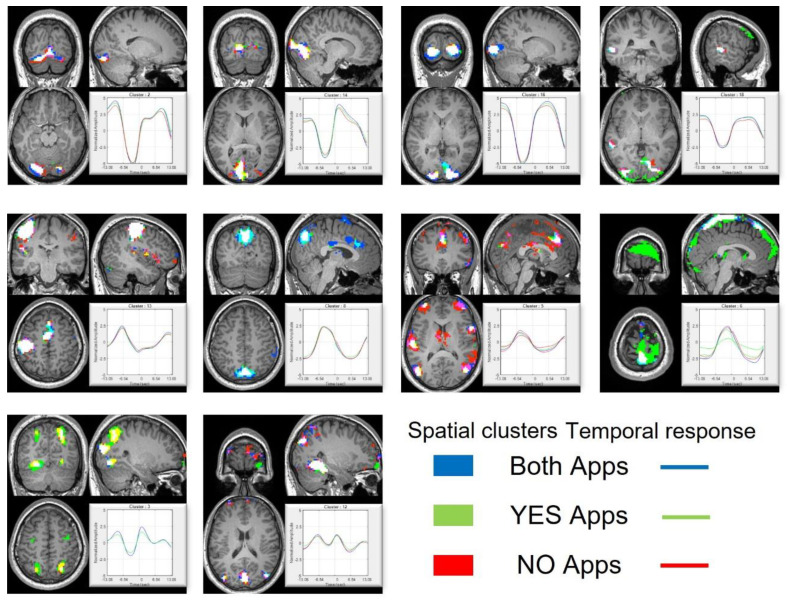
Spatial and temporal results for k-means clustering. Figure shows some of the clusters with meaningful temporal and spatial response. The response for ‘yes’ apps is shown in green, ‘no’ apps is shown in red and the case for all user decision (either yes or no) is shown in blue.

## Data Availability

The datasets generated for this study and the MATLAB codes used for the analysis of the data are available on request to the corresponding author. Requests to access these datasets should be directed to HP, harshit.parmar@ttu.edu.

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
