# Peer review of "Visualization of the Dynamic Brain Activation Pattern during a Decision-Making Task"

_brainsci, 2022, doi:10.3390/brainsci12111468_

Round 1

Reviewer 1 Report

This study aimed to investigate dynamic brain activation pattern during a decision-making task. I have the following suggestions.

1.    What is the novelty of this study although several studies of dynamic brain activation pattern during a decision-making task have been investigated earlier?

2.    Please write down the contribution of the study at the end part of the Introduction section in bulleted form.

3.    The abstract should be improved by combining the objectives, short methodology, main findings results, and prospective application.

4.    Manuscript is difficult to follow and need improvement in writing and drawing figures.

5.    No subject demographics were given.

6.    Authors should add a figure of the experimental protocol and image of real experiment scenario in this study.

7.    Authors should include conceptual figure of their proposed approach with more details and parametrization.

8.    Authors should introduce the mental workload, cognitive impairment using other methods, such as, 2-D brain imaging EEG. Machine-learning approaches are utilized for stroke prediction in article, healthsos: real-time health monitoring system for stroke prognostics; in article, quantitative evaluation of task-induced neurological outcome after stroke; in article, quantifying physiological biomarkers of a microwave brain stimulation device; in article, quantitative evaluation of eeg-biomarkers for prediction of sleep stages; and in article, driving-induced neurological biomarkers in an advanced driver-assistance system.

9.    Quantification of dynamic brain activation pattern during a decision-making task is quite complex and likely influenced by many pathophysiological factors. Due to the very limited data, the results provided in this paper might not support the hypothesis sufficiently.

10.  Authors should describe more details of the MRI data pre-processing, MRI feature extraction and data analysis.

11.  Results and discussion section need to be extended and improved. Authors must make discussion on the advantages and drawbacks of their proposed method with other studies by adding a discussion section.

12.  From the writing point of view, the manuscript must be checked for typos and the grammatical issues should be improved.

Author Response

Thank you for dedicating your time and efforts in providing your valuable feedback on the manuscript. I have incorporated the changes to reflect all the suggestions. All the modifications have been highlighted in yellow in the manuscript resubmission.

This study aimed to investigate dynamic brain activation pattern during a decision-making task. I have the following suggestions.

  1. What is the novelty of this study although several studies of dynamic brain activation pattern during a decision-making task have been investigated earlier?

The novelty of this article is the development and validation of an algorithm for estimation of ‘voxel-wise’ dynamic brain activation pattern. Some previous fMRI studies have looked at dynamic activation but they either use it with GLM framework, which works best with block related experiment design or use sparse L1 regularization, which is an iterative process. To use L1 regularization for whole brain data is not practically feasible. There are some studies which use resting state dynamic functional connectivity (dFC) but again those are limited to a handful of brain regions and a number of time windows. The proposed approach has the ability to estimate the dynamic brain activation independently for each voxel and across the entire time series without the need for any event related or block experiment design. Moreover, the approach is based on a closed form solution (as opposed to an iterative process) making it computationally fast and effective for group level analysis. We also showed the application of the proposed algorithm to tasks where determining the exact onset and offset times are not possible. For example, it is impossible to determine the exact instance a decision is made inside the brain and hence very challenging to use such experiment design with the standard GLM framework.

  1. Please write down the contribution of the study at the end part of the Introduction section in bulleted form.

A paragraph has been added in the Introduction section to indicate the contribution of this article.

  1. The abstract should be improved by combining the objectives, short methodology, main findings results, and prospective application.

The abstract has been modified to incorporate the changes.

  1. Manuscript is difficult to follow and need improvement in writing and drawing figures.

A paragraph has been added at the end of the introduction section to briefly describe the contents of each section.

  1. No subject demographics were given.

Subject demographics (including the number of participants, their age, gender and handedness) are given in section 2.2.

“A total of 22 undergraduate students from a large western university participated in this study. The participants received bonus course credits for their participation. There were 9 males and 12 females (one participant choose not to specify their gender) with an average age of the population being 20.86 ± 1.75 years (min = 19, max = 26). 21 out of 22 participants were right hand dominant. The participants were asked for fill out an informed consent and an MRI safety screening form prior to being in the MRI scanner. An institutional review board (IRB) approval was also obtained for the experiment. The participants were also given instructions about the task and practiced the task on a laptop outside the scanner. The practice task had the exact same user interface, but the apps used in the practice were different from the apps used in actual experiment.”

  1. Authors should add a figure of the experimental protocol and image of real experiment scenario in this study.

Figure 2 shows the experiment protocol for a single trial. For different trials, just the app name, screenshot and description changes. Video 1, Figures 4, 5, and 6 shows the results for real experiment scenarios.

  1. Authors should include conceptual figure of their proposed approach with more details and parametrization.

Figure 1 shows the conceptual summary of the proposed approach. More details are given in paragraph 5 and 6 of section 2.1 along with equations 1, 2 and 3.

  1. Authors should introduce the mental workload, cognitive impairment using other methods, such as, 2-D brain imaging EEG. Machine-learning approaches are utilized for stroke prediction in article, healthsos: real-time health monitoring system for stroke prognostics; in article, quantitative evaluation of task-induced neurological outcome after stroke; in article, quantifying physiological biomarkers of a microwave brain stimulation device; in article, quantitative evaluation of eeg-biomarkers for prediction of sleep stages; and in article, driving-induced neurological biomarkers in an advanced driver-assistance system.

We agree that EEG data would provide higher temporal resolution dynamic information but EEG investigation would require a different experimental setup. The current study focuses on fMRI data acquisition and experiment paradigms which are possible within the fMRI framework. The main contribution here is the development and validation of the algorithm to extract and visualize pseudo stimulus from BOLD fMRI data. EEG may have a better temporal resolution but BOLD fMRI or MRI in general provides better spatial resolution as compared to EEG.

  1. Quantification of dynamic brain activation pattern during a decision-making task is quite complex and likely influenced by many pathophysiological factors. Due to the very limited data, the results provided in this paper might not support the hypothesis sufficiently.

We agree that decision making is a very complicated process and can be impacted by various factors. For the decision-making task, the experiment paradigm was designed in a way that it reduces as much bias as possible. However, the main focus of the current article is on the pseudo-stimulus extraction technique which is applied to a simple decision-making task. The results show that the technique is able to extract dynamic brain activation. As a future work, this approach can be used in a separate more dedicated study to identify the causality of events in a decision-making process.

  1. Authors should describe more details of the MRI data pre-processing, MRI feature extraction and data analysis.

The entire fMRI preprocessing and analysis methodology is described in section 2.3.

“The preprocessing is done using SPM 12 toolbox [32] on MATLAB 2020a. The preprocessing includes

For motion correction, rigid body affine transformation is applied to all the functional volumes to align them to the first functional volume. The rigid body transformation accounts for 6 degrees of freedom that includes 3 translation and 3 rotational motions.

After motion correction the functional and anatomical volumes are coregistered. A 3D affine transform with 12 degrees of freedom is used to align the anatomical and mean functional volume.

Next the images are normalized by mapping both anatomical and functional volumes onto the MNI152 brain atlas [33]. The functional volumes were mapped onto the atlas with a spatial resolution of 3mm x 3mm x 3mm while the anatomical volumes were mapped onto the atlas with spatial resolution of 1mm x 1mm x 1mm.

Temporal signal drift was reduced and spatial smoothing was applied to all the functional volumes. Signal drift was estimated and reduced using a Principal Components Analysis based technique [34]. The spatial smoothing was performed using a 3D Gaussian kernel with a full width half maximum (FWHM) of 5 mm.

The normalized volumes are then segmented into gray matter, white matter, cerebrospinal fluid (CSF), skull and skin. The segmented gray matter, white matter and CSF volumes are used to obtain a brain mask. Any voxels outside the brain region were discarded from further analysis to reduce the amount of data and computation time.

Before explaining each component of the estimation, we will lay out the overall process.

  • Extract the pseudo stimulus for each voxel
  • Normalize the response times
  • Average together all of a single participants response level across all trials.
  • Repeat for only the trials with a yes and only the trials with a no answer.
  • Use clustering to compress the 50,000+ voxels’ responses into 20 clusters.

The preprocessed data is then used to extract the pseudo-stimulus for each participant. First the entire 4D fMRI volume is converted to a 2D matrix of size T x Nv. Nv is the total number of voxels that lie within the brain region and T is total number of brain volumes in the data. Each column of this matrix corresponds to that voxel’s time series. This is the matrix ‘Y’ as described in section 2.1. From the matrix ‘Y’, the weight matrix is estimated which is of size N x Nv, where N corresponds to the number of regressors in the design matrix. The first ‘T’ rows of the weight matrix correspond to the pseudo stimulus estimate for each of the Nv voxels. The voxel-wise pseudo-stimulus is estimated for all the participants.

The response times varied across apps and across participants thus a one-to-one comparison is not possible. All the stimuli where participants responded before 10 seconds were discarded. The average number of stimuli that were discarded per participant was 5.3 (min = 0; max = 16; median = 4). Thus, it can be assumed that all the responses were at least 10 seconds long and with 5 seconds blanking period making a total time of 15 seconds. To have uniformity in analysis, the onset of the stimulus is considered as the reference point and 13 seconds (25 TRs, including the reference point) before and after the reference point are considered for each app. Thus, for each stimulus, a fixed length window is obtained which is centered at the onset of stimulus. This fixed window allows us to compare the response across different apps and participants. For each participant, the pseudo-stimulus in the fixed window across all the apps is averaged together to obtain a participant level response to the stimulus.

For each participant, the pseudo-stimulus is used to obtain the participant level response to the app download decision. The participant level response matrix is of size 49 x Nv (the reference point being the 25th row). Let’s assume that the kth trial began at time instance tk. Corresponding to that, the time window would be [tk – (24) • TR] to [tk + (24) • TR]; for k = {1, 2, 3, …, 50}. For a given participant, all the 49 x Nv matrices (corresponding to each value of k) are averaged together to have single participant level response to the app.

The participant level response shows how the brain activity of the participant changed when responding to the app download decision. Three separate responses were obtained for each participant, one for all the apps, one for apps for which the participant decided to download (YES apps) and the last one for the apps the participant didn’t decide to download (NO apps).

Finally, considering the response for each voxel as features, the voxels are clustered into 20 different groups using the k-means clustering approach. For each cluster, a representative response is obtained by grouping the voxel response for all the voxels belonging to same cluster. At the end of clustering the 49 x Nv matrix is converted to a 49 x 20 matrix with each column containing the temporal response to a single cluster.

The clustering and representative cluster response was obtained for all participants. However, the cluster assignment for k-means clustering is randomized and a correlation-based post processing step was used to match the clusters across all participants. The group level analysis was performed by combining the spatial clusters and cluster response for all participants. Useful information can be extracted from the spatial clusters and cluster-wise response.”

Moreover, the entire dataset and the MATLAB code used to generate the results are available upon request for anyone who wish to replicate the results shown in the article.

  1. Results and discussion section need to be extended and improved. Authors must make discussion on the advantages and drawbacks of their proposed method with other studies by adding a discussion section.

A paragraph has been added to the ‘Results and Discussions' section to indicate the advantages offered by the proposed technique. A comparison of the proposed approach with other deconvolution approach is also shown in the appendix section.

The limitation of the proposed technique along with some future work is already discussed in the last two paragraphs of the ‘Results and Discussion’ section.

  1. From the writing point of view, the manuscript must be checked for typos and the grammatical issues should be improved.

The article was checked both manually and with ‘Spelling and Grammar’ check feature of Microsoft Word. All indicated spelling and grammatical errors were corrected.  We welcome your input on any specific errors that might have led to this comment.

Reviewer 2 Report

Figure 1 is not clear to me. Labels A, B, C, D, E, F are not correctly placed. Color plots in figure 1 are barely visible.

Please define TR (repetition time) in the text as well, apart from figure 1.

How is pseudostimulus normalized (line 330)?

The meaning of “perceived input” in line 348 is not clear. Do the authors mean projected input of the pseudostimulus?

In figure 6, it is not clear to me what is ‘both yes and no.’ Does it refer to when participants indicate both choices by pressing both buttons?

Author Response

Thank you for dedicating your time and efforts in providing your valuable feedback on the manuscript. I have incorporated the changes to reflect all the suggestions. All the modifications have been highlighted in yellow in the manuscript resubmission.

Figure 1 is not clear to me. Labels A, B, C, D, E, F are not correctly placed. Color plots in figure 1 are barely visible.

Figure 1 has been modified to include all the changes. The color plots are made thicker so that they are clearly visible. Labels have also been placed correctly along with label indicator with each subplot.

Please define TR (repetition time) in the text as well, apart from figure 1.

TR has been defined in the text in lines 115-116.

How is pseudostimulus normalized (line 330)?

The estimated weights are normalized so that they have zero mean and unit standard deviation (across space and time). Once the estimated pseudostimulus is averaged across all trials for a given participant, there will be N*T unique weights where ‘N’ is the number of voxels inside the brain and ‘T’ is number of time points in a trial. [The trial consists of 25 TRs before and 25 TRs after the beginning of the new stimulus, as described in paper. Thus T = 49]. For these N*T weights, mean (M) and standard deviation (S) are computed, and all the weight values are normalized with respect to that. NWi = (Wi – M)/S; where Wi is the ith weight and NWi is the corresponding ith normalized weight. 

The meaning of “perceived input” in line 348 is not clear. Do the authors mean projected input of the pseudostimulus?

Yes, perceived input does mean input of the pseudo stimulus to the brain which causes the observed BOLD response. A description of that has been added to section 2.1 (lines 106 – 108).

In figure 6, it is not clear to me what is ‘both yes and no.’ Does it refer to when participants indicate both choices by pressing both buttons?

The ‘YES apps’ refer to the apps for which the user indicated to download the app. Similarly, ‘NO apps’ refers to the apps for which the user indicated to not download the app. By ‘BOTH’ we meant to convey all the apps irrespective of their decision. I agree it was creating confusion. Thus, we have modified Figure 6 and its caption to ‘ALL’ apps instead to avoid any further confusion. The user can only respond either YES or NO and not select both.

Round 2

Reviewer 1 Report

Most of the review comments are not addressed. The manuscript was not improved. 

The novelty of this study is not clear.

This study lacks MRI domain-specific knowledge. 

Author Response

Thank you for your very quick response to the modified version of the manuscript. We uploaded a modified version of the manuscript along with a point-by-point response to all of your previous comments earlier. While we appreciate your feedback, we respectfully disagree with your current comments. Many of your comments were about adding things that were already there in the initial submission of the manuscript e.g. Participant demographics. Below is a point-by-point response to your current comments (comments highlighted in BOLD followed by our response). We believe content related to current comments are already included in the latest version of the manuscript.

Most of the review comments are not addressed. The manuscript was not improved. 

All the comments were addressed and a point-by-point response was also attached to the modified manuscript.

The novelty of this study is not clear.

The abstract, introduction, and discussion sections were modified to include the novelty of the study in the modified version submitted previously.

This study lacks MRI domain-specific knowledge. 

The entire study is about functional MRI (and not EEG) and all the stages of the experiment starting from Experiment design to data processing are described in Section 2 of the manuscript.